# Chitosan–Azide Nanoparticle Coating as a Degradation Barrier in Multilayered Polyelectrolyte Drug Delivery Systems

**DOI:** 10.3390/biom9100573

**Published:** 2019-10-05

**Authors:** Steffen Sydow, Armin Aniol, Christoph Hadler, Henning Menzel

**Affiliations:** Institute for Technical Chemistry, Braunschweig University of Technology, 38106 Braunschweig, Germany; s.sydow@tu-braunschweig.de (S.S.); a.aniol@tu-braunschweig.de (A.A.); ch.hadler@tu-braunschweig.de (C.H.)

**Keywords:** Layer-by-Layer (LbL), nanoparticles, chitosan

## Abstract

Therapeutics, proteins or drugs, can be encapsulated into multilayer systems prepared from chitosan (CS)/tripolyphosphat (TPP) nanogels and polyanions. Such multilayers can be built-up by Layer-by-Layer (LbL) deposition. For use as drug-releasing implant coating, these multilayers must meet high requirements in terms of stability. Therefore, photochemically crosslinkable chitosan arylazide (CS–Az) was synthesized and nanoparticles were generated by ionotropic gelation with TPP. The particles were characterized with regard to particle size and stability and were used to form the top-layer in multilayer films consisting of CS–TPP and three different polysaccharides as polyanions, namely alginate, chondroitin sulfate or hyaluronic acid, respectively. Subsequently, photo-crosslinking was performed by irradiation with UV light. The stability of these films was investigated under physiological conditions and the influence of the blocking layer on layer thickness was investigated by ellipsometry. Furthermore, the polyanion and the degree of acetylation (DA) of chitosan were identified as additional parameters that influence the film structure and stability. Multilayer systems blocked with the photo-crosslinked chitosan arylazide showed enhanced stability against degradation.

## 1. Introduction

The development of tailored drug delivery systems that can release a variety of novel therapeutics, proteins or drugs is a mayor challenge in the improvement of targeted tissue engineering applications [1,2]. For example, the preparation of cell instructive implants requires spatiotemporal control of more than one signaling protein [3]. Nanogel-based protein release systems combined with the Layer-by-Layer technique (LbL) have been proven to be a good compromise between spatial release control and easy scalability of the process [4]. Furthermore, it was shown that multilayer films prepared by LbL can be constructed easily by spray- or dip-coating processes [5]. It was possible to prepare release systems for a variety of drugs, e.g., ampicillin [6], siRNA [7] or the antitumor drug, doxorubicin [8]. Photochemically reactive biopolymers were also reported as potential drug delivery matrices, scaffolds in numerous tissue-engineering applications and for drug- and cell delivery into tissues [9,10,11]. The use of additionally crosslinked hydrogels can retard the diffusion of encapsulated proteins or drugs. Furthermore, the degradation behavior of hydrogels was improved or the bioactivity was enhanced by photochemical crosslinking [11,12].

Recently, we showed the construction of nanogel based multilayers systems [4]. The biopolymer chitosan (CS) was used to prepare nanogel particles by ionotropic gelation with tripolyphosphate (TPP). These nanoparticles were shown to have a defined particle size and a positive zeta-potential [3,13,14]. When adsorbed to surfaces, these nanoparticles reorganize into homogenous films [15]. Using the LbL technique and polyanions alginate (Alg), chondroitin sulfate (Cho) or hyaluronic acid (HA), respectively, multilayered films can be prepared with CS–TPP. The behavior of these multilayer films depended on pH-value, dipping time, and ionic strength of the polyelectrolytes [16,17]. The stability of these multilayer films, however, has not yet been addressed in detail, although it is very important for an application as a drug-releasing implant coating. An uncontrolled release of drugs, as would take place if the coating is not stable or if the drug diffusion is too fast, must be prevented. 

In this study, we synthesized photochemically crosslinkable chitosan aryl azide (CS–Az) using the coupling agent, dimethoxyl-1,3,5-triazin-2-yl)-4-methylmorpholinium chloride (DMT-MM). Nanoparticles were prepared by ionotropic gelation of the CS–Az with TPP and characterized in terms of particle size, zeta potential and long-term stability. Multilayer systems were prepared from CS–TPP nanoparticle suspensions and alginate, chondroitin sulfate or hyaluronic acid by the LbL technique and CS–Az–TPP nanoparticles were used to prepare a blocking layer on top of the multilayer system. 

## 2. Materials and Methods

### 2.1. Materials

Chitosan (190–310 kDa, degree of deacetylation (DDA): 75–85%), acetic anhydride (>99%), dimethoxyl-1,3,5-triazin-2-yl)-4-methylmorpholinium chloride (DMT-MM, >96%), alginic acid sodium salt (70–80 kDa) and hyaluronic acid (15–18 kDa) were purchased from Sigma-Aldrich (St. Louis, MO, USA). 4-Azidobenzoic acid (96%) was obtained from TCI, Ltd. (Tokyo, Japan), and chondroitin sulfate (90%) from Alfa Aesar (Haverhill, MA, USA).

### 2.2. Purification and Adjustment of the Degree of Acetylation (DA)

Before acetylation, chitosan was purified by the method according to Gan et al. [18]. 

To adjust the degree of acetylation, chitosan was reacted with acetic anhydride. To do so, purified chitosan (DA: 17% as determined by NMR) was dissolved in 1% (*v*/*v*) acetic acid to a concentration of 5 mg/mL. After the chitosan was dissolved completely, the volume of chitosan solution was doubled with the addition of ethanol. Acetic anhydride was added to the solution in an amount necessary for the desired DA and calculated according to Freier et al. [19]. The mixture was stirred for 18 h at room temperature. The acetylated chitosan was dialyzed against deionized water and subsequently lyophilized. The characterization by ^1^H-NMR indicates a DA of 42%. 

### 2.3. Synthesis of Chitosan-Aryl Azide

For synthesis of Chitosan-aryl azide (CS–Az), a modified protocol of that described by Hadler et al. was applied [20]. Purified chitosan (DA 17%) was dissolved in 0.005 M HCL solution. The solution was stirred until the chitosan was dissolved completely. Sodium 4-azidobenzoate (in 0.005 M HCL) was added to the chitosan solution in different molar ratios calculated to the chitosan amino groups (0.05–0.16 mmol). The pH-value of the mixture was adjusted to pH = 6.0 using 1 M NaOH. After adding the coupling reagent DMT-MM (1.2-fold excess), the mixture was stirred at room temperature under exclusion of light for 48 h. The resulting CS–Az was purified by dialysis against 0.1 M NaCl and deionized water and subsequently lyophilized. The degree of modification was determined by ^1^H-NMR spectroscopy. 

### 2.4. Multilayer Formation with Alginate (Alg), Chondroitin Sulfate (Cho) or Hyaluronic Acid (HA)

CS–TPP nanoparticles were prepared by dissolving chitosan (DA 17% or DA 42%) in 0.1% (*v*/*v*) AcOH to reach a concentration of 2 mg/mL. In addition, TPP was dissolved in Millipore water to yield a 2 mg/mL solution. The TPP solution was rapidly mixed with the CS solution in a ratio of 3:1 (CS–TPP), resulting in defined nanoparticles. Particle size and zeta potential were analyzed immediately. Multilayer coatings have been prepared on silicon substrates (Microchemicals, Ulm, Germany) by the previously published method [4]. CS–TPP nanoparticle suspensions were used with a concentration of 2 mg/mL. The anionic biopolymers were dissolved in Millipore water to reach a concentration of 5 mg/mL. A dip robot (Riegler and Kirstein, Berlin, Germany) was used for the alternating dipping for the Layer-by-Layer method:

*Nanoparticle coating*: The substrate was dipped into the nanoparticle suspension for 10 min, followed by a short drying step in a N2 stream. After washing in 0.1% acetic acid and deionized water for 1 min each, the layer was dried in a N2 stream. 

*Interlayer coating:* The nanoparticle-coated silicon substrate was dipped into the solution of Alg, Cho or HA for 10 min. After drying in a N_2_ stream, the interlayer was washed with deionized water for 1 min, followed by another drying step.

### 2.5. Preparation of CS–Az “Blocking Layer”

CS–Az (1 mg/mL) was dissolved in 0.1% AcOH and was stored in the dark until use. CS–Az–TPP nanoparticles were prepared by the same procedure as the CS–TPP nanoparticles (see above). After preparation of the multilayers on silicon substrates as described above, they were dipped into the CS–Az–TPP nanoparticle suspension for 10 min, followed by a short drying step in a nitrogen stream. After consecutive washing steps in 0.1% AcOH and deionized water for 1 min, a second drying step in a N_2_ stream was carried out. For photochemical crosslinking, a mercury vapor lamp (LOT-ORIEL, Germany) with a UG11 glass filter (SCHOTT, Mainz, Germany) was used. The radiation intensity of the UV-light (250–400 nm) was determined to be 100 mW/cm^2^. Crosslinking was carried out by irradiation for 5 min. 

### 2.6. Degradation Studies for CS–Az Blocked Multilayers

Degradation studies were carried out by immersion of the coated wafers in PBS-buffer at 37 °C. Three samples were tested for each of the following groups. One group was coated with photochemically crosslinked CS–Az–TPP and lysozyme (1.5 μg/mL) was added. The second group had no CS–Az–TPP top layer, and lysozyme was added. The third group comprised coating without the CS–AZ–TPP top layer and no added lysozyme. At defined time points, the resulting dry layer thickness was analyzed using ellipsometry. 

### 2.7. Instruments

^1^H-NMR spectra were recorded on a BRUKER AV II-300 (300 MHz) at 25 °C using D_2_O/DCl 285:1 or 500:1 as solvent with DSS (4,4-dimethyl-4-silapentane-1-sulfonic acid) as an internal reference. NMR-spectra were analyzed by *MeReNova* 9.0. FTIR spectra were recorded on a Bruker Equinox 55 spectrometer. A mercury cadmium telluride detector and a ZnSe crystal ATR accessory (Harrick Scientific Products Inc., Pleasantville, NY, USA) were used for detection. Particles size and zeta potential were measured with a Zetasizer Nano ZS from *Malvern Instruments* (Malvern, UK). Size measurements were carried out in disposable sizing cuvettes (DTS0012) and for zeta potential measurements, disposable folded capillary cells (DTS1070) were used. The measurements were carried out at 20 °C. For analysis, Malvern Zetasizer Software Version 7.03 was used. For all ellipsometric measurements, a Multiscope (*Optrel*, Sinzing, Germany) was used. Each silicon wafer was measured as uncoated sample before use as a reference. Measurements were carried out in x,y-mode at an angle of 70°. The surface of the coated wafers was scanned with 16 data points in a 4x4 matrix. Data evaluation was carried out using ELLI Version 3.2 from Optrel.

## 3. Results and Discussion

### 3.1. Polymer Synthesis

Photochemically crosslinkable chitosan was obtained by modification with arylazide groups under mild conditions by using 4-azidobenzoic acid and dimethoxyl-1,3,5-triazin-2-yl)-4-methylmorpholinium chloride (DMT-MM) as a coupling reagent. The number of amino groups along the chitosan chain depends on the degree of acetylation (DA), which was DA = 17% and corresponds to 83% amino groups for the commercially available chitosan used for the modification. The use of DMT-MM offers the opportunity to carry out the reaction directly in water under neutral pH conditions without any organic solvents, additives, or the need to adjust the pH with buffers [21,22]. For all functionalization reactions, excess of DMT-MM (1.2 fold) was used to ensure full activation of the 4-azidobenzoic acid. The successful functionalization of chitosan was proven by the appearance of two new aromatic peaks at 7.84 and 7.22 ppm in the ^1^H-NMR spectra (see Figure 1). This is in line with the results of Hadler et al. [20]. Using these proton signals, the degree of substitution (DS) was determined to be 3.3%. CS–Az with a higher degree of substitution of 6.8% was also prepared, but the particles obtained with these polymers were highly disperse in size and unstable (see Appendix A, Appendix A). FTIR-spectra (see Figure 2) of all chitosan derivatives show broad peaks from 3250 – 3500 cm^−1^ originating from O-H and N-H stretching vibrations, peaks at 1620 cm^−1^ and 1510 cm^−1^ can be assigned to C=O stretching (amide I) and N-H deformation vibrations (amide II). For the CS–Az (see Figure 2b), an additional peak appears at 2120 cm^−1^, which can be attributed to the N_3_ stretching vibration of the arylazide [20,23,24]. After irradiation with UV-light for 5 min, this azide peak at 2120 cm^−1^ is no longer present (see Figure 2c). This indicates full conversion of the azide in the photoreaction. 

### 3.2. Preparation of Nanoparticles

Chitosan nanoparticles have been described for a broad range of drug delivery applications [13,14,18,19,25,26]. Even photo-crosslinkable chitosan particles and hydrogels have already been established as potential delivery systems [27,28,29]. CS–TPP nanoparticles were prepared by ionotropic gelation by adding tripolyphosphate anions (TPP) to a chitosan (CS) solution. They were characterized by dynamic light scattering in terms of particle size and zeta potential (see Figure 3). Nanoparticles prepared from non-modified chitosan CS(17)–TPP showed a monomodal size distribution with a particle diameter of 190 ± 2 nm, which is in line with previous publications dealing with CS–TPP nanoparticles [4,13]. Employing the azide-modified chitosan (CS–Az) in the ionotropic gelation with TPP, the particle size of the nanoparticles obtained decreases to 114 ± 4 nm. In contrast to a smaller particle size, the size distribution broadens. Both might be a consequence of the hydrophobic aryl azide groups, which should result in some shrinkage of the polymer coils in water. CS–A with a higher degree of substitution with azide-groups, resulted in increasing particle size and highly disperse size distribution (see Appendix A, Appendix A) and the particle stability decreased. Therefore, all further investigations were carried out with CS–Az with 3.3% azide groups.

For photo-crosslinking, the CS–Az nanoparticles were irradiated at 100 mW/cm^2^ for 5 and 10 min using a mercury vapor lamp. To obtain UV-light in the required wavelength range, an optical glass filter was used—330 nm ± 70 nm. The particle size was measured again immediately after the irradiation process. No significant change in the particle size distribution or polydispersity index (PDI) was found (see Figure 3). After 5 min of irradiation, the particle size was 109 ± 5 nm, while after 10 min, a particle size of 119 ± 6 nm was measured. Even long-term particle size measurements did not show a significant change in size distribution over a time of one week, as shown in Figure 4.

The zeta-potential of arylazide-modified CS–Az–TPP nanoparticles was difficult to determine because of some tendency for sedimentation of the particles during the measurements. Nevertheless, DLS measurements indicated a positive zeta potential of approximately +16 mV. After crosslinking with UV-light, the zeta potential did not change significantly. A deposition of positively charged CS–Az–TPP nanoparticles on a negative surface should, be possible as it was shown before with non-modified CS–TPP nanoparticles.

### 3.3. Degradation Studies Nanoparticles

Chitosan nanoparticle stability is a crucial parameter for use in medical applications. The polysaccharide is very stable; however, enzymatically degradable, e.g. by lysozyme [30]. This enzyme is present in human serum [31] and can degrade the chitosan. This enzymatic degradation results from cleavage of the 1,4-beta-linkage between the N-acetyl-D-glucosamine and D-glucosamine. The degree of acetylation (DA) is important for the degradation rate. An optimum degradation rate was shown in a range of DA = 30–70% [19,32,33]. CS(42) has a high DA and is expected to be degraded faster compared to CS(17) [15,19]. The UV-crosslinked CS–Az particles need to be stable in the presence of lysozyme to use them as retarding elements in a coating. To prove this, degradation studies were carried out. For all CS–TPP nanoparticles, the degradation studies were performed in the absence or presence of 1.5 µg/mL of lysozyme at 37 °C, which is approximately the enzyme concentration in human serum [31]. The particle size was then measured after different times and any changes in particle size indicated degradation. 

CS(17)–TPP nanoparticles without arylazide modification were stable over a period of 21 d if no lysozyme was present, as shown in Figure 4. Upon addition of lysozyme, the particle size changed significantly to 400 nm after 1.5 days of incubation. This increase in particle size was an indication that the nanoparticles lost stability and larger aggregates are formed. Such a behaviour has been observed before upon degradation of the CS–TPP nanoparticles [34] and can be explained by a different ratio of TPP to Chitosan upon degradation of the chitosan. It had been shown before that the ratio of Chitosan to TPP strongly influenced the size of the particles and that there is a minimum particle size [35]. Upon degradation of the Chitosan, the ratio between TPP and Chitosan was no longer optimal and larger particles resulted. Upon longer degradation times, no more particles were observed, indicating the extensive degradation of the chitosan. On the other hand, the modified and crosslinked CS–Az–TPP nanoparticles were stable over the three-week measurement time, showing no changes in particles size (122 nm). Even after the addition of lysozyme, particle size did not change significantly. After three weeks, only minor changes in particle size (117 nm) were observed. This indicated that the degradation by lysozyme is blocked or at least retarded very strongly by the modification and photo-crosslinking. Therefore, CS–Az–TPP nanoparticles appeared to be interesting for use in the preparation of blocking layers to stabilize multilayer coatings.

### 3.4. Degradation Study for CS–Az Blocked Multilayers

Drug-releasing coatings can be prepared via Layer-by-Layer deposition of the positively charged CS–TPP nanoparticles and polyanions [4,13,36]. The method of multilayer preparation is easily scalable and can produce coatings with gradients of different drugs and, therefore, is suited for targeted release [4]. Nevertheless, the multilayer systems must meet the stability requirements. 

As described previously, the chitosan-based multilayer systems are degradable. The solubility of chitosan in water depends on the degree of acetylation [33]. The stability of the nanoparticle films against degradation and dissolution can be adjusted by tailoring the DA. LbL-multilayers prepared from nanoparticles with chitosan having a high degree of acetylation (DA = 42) were expected to degrade under physiological conditions [33]. To counteract this effect, photo-crosslinkable CS–Az–TPP nanoparticles were tested to prepare a “blocking layer” at different positions to retard degradation of the multilayers. Scheme 1 shows the method of preparation of the multilayers and the addition of the azide “blocking layer” in detail. As shown above, CS–Az–TPP nanoparticles were not degraded even when lysozyme was added, because of their crosslinked character (Figure 4). Here, we investigated the degradation of the multilayer by monitoring the decrease of dry thickness of the films after immersion in phosphate-buffered solution (PBS).

The multilayers were prepared by dipping silicon alternately into CS–TPP nanoparticle suspensions and a solution of alginate, chondroitin sulfate or hyaluronic acid, respectively. Employing different polyanions resulted in multilayers with different internal structures [4]. In general, weak polyelectrolytes are highly coiled, compared to strong polyelectrolytes, which are more elongated. Therefore, weak polyelectrolytes result in thicker multilayers (Table 1) and stronger penetration of the counter-ions into the layers [4]. Chitosan typically also behaves as a weak polyelectrolyte and can interact differently, depending on the acid strength of the polyanions [37,38,39,40,41].

It was found previously, that upon deposition of the polyanion layers on the CS–TPP film, TPP was released and the system underwent significant structural changes [42,43]. XPS measurements showed that no more TPP is left in a CS–TPP/Alg double-layer [4]. SEM images of such films show a homogeneous layer without remarkable topographical features [4,13]. Replacement of TPP in the particle film by the polyanions should influence the film stability; it is suspected, that the stability of the film is reduced compared to the nanoparticle in suspension. To investigate whether and to what extent a photo-crosslinked blocking layer can stabilize the film; CS–Az–TPP was coated atop the multilayer (11th layer). The degradation of the films was monitored by measurements of the dry layer thickness of the multilayers by ellipsometry at different time points. The layer thickness of a single CS–Az–TPP layer on a Si-wafer was determined, separately, to be 16 ± 5 nm. For the multilayer systems, the thickness of the CS–Az–TPP top layer was calculated as the difference between the 10^th^ layer and complete film with the CS–Az–TPP (11 layers), as shown in Table 1. The layer thickness of a single CS–Az–TPP layer is comparable to a single layer of adsorbed CS–TPP nanoparticles, as measured previously [4]. 

Three experimental groups were investigated. In all three groups, systems consisting of 10 layers of CS–TPP with Alg, Cho or HA, respectively, were used. In the first group, degradation is investigated without adding any lysozyme (control w/o Lys). In the second group, the degradation experiments were carried out by adding lysozyme to the incubation solution (with Lys). The concentration of lysozyme was adjusted to be in accordance with the human serum concentration (~1.5 µg/mL) [31]. In the third group, a crosslinked CS–Az–TPP top coating was applied and the degradation was performed in the presence of lysozyme (blocked and Lys added).

In the case of the CS(42)–TPP/Alg system, there is a strong initial decrease of the layer thickness of the multilayer coating upon incubation with PBS buffer, as shown in Figure 5a. After 1 day of incubation, the multilayer thickness had decreased to 30% of the original thickness, even without adding lysozyme. Thus, the CS(42)–TPP/Alg coating was not stable against buffer at physiological conditions, most likely because of the high water solubility of CS(42) [33]. In contrast to the low stability as a coating, Poth et al. showed that CS(42)–TPP particles in suspension are stable at room temperature for more than 2 weeks [15]. This CS–TPP nanoparticle stability was confirmed by dynamic light scattering measurements, which showed no relevant change in particle size and PDI after 10 d [4]. However, in the coating, the TPP counter ions have been replaced completely by the alginate polyanions [4], which seemed to reduce the stability significantly.

Adding Lysozyme accelerated the CS(42)–TPP/Alg film degradation further and resulted in lower layer thickness (Alg with Lys, see Figure 5a). When a CS–Az–TPP layer is added as a top coating, again a degradation was observed as a reduction of the multilayer film thickness (Alg blocked with CS–Az, see Figure 5a). However, 70% of the layer thickness was left after 1 day, and approximately 40% of the initial dry film thickness after 30 days of incubation. This was approximately twice as high as in the case of the non-blocked multilayer systems.

CS(17) has a reduced water solubility compared to CS(42) [15,33]. Therefore, a significant slower reduction of layer thickness was expected for a CS(17)–TPP/Alg film. However, a fast initial reduction within 1 day was also observed in this case (Figure 5b), although it is less pronounced (58% remains, compared to 30%) compared for CS(42)–TPP/Alg film. Again, the reduced stability of the film compared to the nanoparticles in suspension (see degradation studies on the nanoparticles) was rationalized with the exchange of the TPP counter ions by alginate. 

Addition of lysozyme to the buffer caused CS(17)–TPP/Alg films to decrease to 49% of the original film thickness within one day. Viscosity measurements performed by Poth have shown that degradation of CS(17)–TPP nanoparticles took place rapidly—within one day after adding 1.5 µg/mL lysozyme [15]—confirming other reports on the degradability of low acetylated chitosan [19,30,44,45]. Therefore, fast reduction of the layer thickness was attributed to degradation of chitosan by lysozyme. Using CS–Az–TPP nanoparticles as coating layer atop the CS(17)–TPP/Alg multilayers again slowed down the degradation; with this blocking layer and lysozyme, the same degradation kinetics was observed as w/o lysozyme and reached 39% after 40 days. Comparing the effect of the blocking layer between the multilayer with different DA (Figure 5a,b), it could be stated that with the blocking layer, similar degradation kinetics were observed for both CS(17) and CS(42). 

Additionally, CS–Az–TPP particles were used to build up an intermediate layer (5th layer). The intermediate blocking layer was crosslinked by UV-irradiation before adding additional layers. However, the degradation kinetics did not change significantly compared to the control w/o lys (see Appendix A, Appendix A). After the large initial reduction, the breakdown of the multilayer slows down after an incubation time of 40 d, at which 50% (top-blocked), 32% (intermediate-blocked) and 24% (non-blocked) of the original layer thickness are left, respectively. Thus, compared to the top-blocked system, an intermediate CS–Az–TPP layer has less influence on the degradation.

Compared to alginate (Alg), chondroitin sulfate (Cho) bearing a sulfate group is a stronger polyelectrolyte [46]. We supposed that the higher ionic strength of Cho results in more deposited polymer chains and thus a higher multilayer thickness compared to Alg [4]. By full exchange of the TPP during the film deposition, in combination with the higher multilayer film thickness, the Cho systems should be less stable against aqueous conditions. The multilayers resulting from CS(17)–TPP with Cho showed a similar reduction of the layer thickness in all experimental groups (i.e., w/o lysozyme, with lysozyme, with lysozyme plus blocking layer) (Figure 5d). There was a strong initial reduction of the multilayer film thickness for Cho with Lys and Cho w/o lysozyme. Because of the strong initial thickness reduction in both groups, it was deduced that the degradation by lysozyme is less important compared to dissolution of the layer in the buffer. Adding a CS–Az–TPP top layer did not significantly influence the stability and the film thickness reduction was not retarded. However, all multilayers with Cs(17)–TPP and Cho as polyanions show a greater layer thickness after incubation compared to the Alg multilayers.

The CS(42)/Cho multilayers behaved differently, as shown in Figure 5c. A fast initial reduction—within 1 day—was observed for Cs(42)–TPP/Cho without lysozyme to 50%; however, the layer thickness remained constant. Therefore, a higher layer thickness results after 1 day of incubation compared to Alg. However, with the addition of lysozyme, a strong reduction to only 18% of the original layer thickness after 1 day was observed. This can be attributed to the better degradability of CS(42). Adding CS–Az–TPP as the top layer retards the degradation/dissolution of the multilayer system and a layer thickness of 42% after 1 day was observed. 

In contrast to chondroitin sulfate (Cho), hyaluronic acid (HA) behaves typically as a very weak polyanion [39,47]. Since chitosan is also a weak polyelectrolyte, the resulting films have a high thickness and show an exponential thickness growth, as reported previously [4]. Therefore, it was expected that the stability of the layer against aqueous conditions is low [48,49]. Indeed, HA multilayer systems showed a high burst degradation for CS(17)–TPP/HA, as well as for CS(42)–TPP/HA (see Figure 5e,f). The multilayers with higher acetylated CS(42)–TPP/HA were reduced to only 7% of the original layer thickness afte r 1 day of incubation (Figure 5e). In the case of CS(17)–TPP/HA, a similar decrease of the layer thickness to 20% of the initial thickness was observed (Figure 5f). No significant influence of lysozyme or the CS–Az–TPP blocking layer was noticeable. 

The different polyanions resulted in different internal structures of the polyelectrolytes multilayers and different layer thickness. These differences in the internal structures also influenced the film stability upon immersion into buffer under physiological conditions. The less compact CS–TPP/HA is degraded much faster than the more compact CS–TPP/Cho or CS–TPP/Alg films. In addition, these differences might have an influence on a potential drug release from the multilayers. Therefore, an incorporated drug might be released from CS–TPP/HA multilayers significantly faster. In the case of the less compact CS(42)–TPP/HA, a CS–Az–TPP blocking layer has no influence on the degradation rate. However, in multilayer films with Cho as polyanions, a retardation of the film degradation was observed due to the CS–Az–TPP blocking layer. This retardation is even stronger in the case of Alg as polyanions in the multilayers. Because of the reduced water solubility of CS(17) compared to CS(42), the influence of CS–Az–TPP on the degradation rate of the multilayer system is lower. It was concluded that CS–Az as a blocking layer could further stabilize the multilayers, which already showed some stability, while the effect on films that are already less stable is negligible.

## 4. Conclusions

Suspensions of CS–TPP nanoparticles were prepared by ionotropic gelation of chitosan (CS) with tripolyphosphate (TPP) and used to build-up multilayer films by Layer-by-Layer deposition using alginate, chondroitin sulfate or hyaluronic acid as polyanions. The type of CS used was varied in terms of the degree of acetylation (DA). The stability of the multilayers was tested by incubation in aqueous buffers under physiological conditions with and without the addition of lysozyme, which can degrade chitosan with intermediate DA. Significant differences were found for the multilayer systems with different polyanions and different DA. The higher the DA, the lower the film stability. Furthermore, the stability decreases from alginate, followed by chondroitin sulfate to hyaluronic acid.

Arylazide-modified Chitosan (CS–Az) with 3.3% of arylazide groups was used to prepare photo-crosslinkable nanoparticles by ionotropic gelation with tripolyphosphate (TPP). By dynamic light scattering, a defined particle size of 114 ± 4 nm and a zeta potential of roughly +16 mV were determined. CS–Az–TPP nanoparticles were deposited atop of the multilayers layers to form a potentially stabilizing layer, which was crosslinked by irradiation. Degradation experiments revealed different effects of the CS–AZ–TPP layer on the film stability. For those films which are relatively stable against degradation, such as CS(17)–TPP/Alginate, the CS–AZ–TPP layer has little effect. However, in films of medium stability, such as those with higher acetylated chitosan CS(42)–TPP/Alginate, a significant stabilizing effect was observed. The multilayers with chondroitinsulfate fall into the same category. However, multilayers with HA showed low stability irrespective of the DA of the chitosan and no stabilization by the CS–Az–TPP coating was observed.

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
