# Peer review of "Chitosan–Azide Nanoparticle Coating as a Degradation Barrier in Multilayered Polyelectrolyte Drug Delivery Systems"

_biomolecules, 2019, doi:10.3390/biom9100573_

Round 1

Reviewer 1 Report

This paper describe a nanoparticles having degradation barrier using photochemically crosslinkable chitosan arylazide (CS-Az). The authors demonstrated the long-term stability of the nanoparticles with the photo-crosslinked CS-Az layer. The multilayered films were prepared on silicon substrates using the nanoparticles and polyanions by a LbL technique, and interestingly, the stability of the films were different depending on polyanions, the degree of acetylation and with/without the CS-Az layer. The insights obtained in this work will be useful for designing DDS nanoparticles. I recommend acceptance for publication in Biomolecules after addressing the following issue.

1) The authors explained that the merits of cross-linking nanoparticles are to retard diffusion of drugs/proteins, and to improve degradation (e.g., P.1, L.34-38). In this paper, they demonstrated the degradation of their materials (nanoparticles and flat films), but they did not study the diffusion of drugs/proteins. The authors should add the drug release data.

2) About nanoparticles. The authors used the CS-Az with 3.3 % Az groups. Can the authors supply more DLS data using the CS-AZ with different Az contents?

3) The authors investigated degradation of CS-Az blocked multilayers prepared on silicon substrates. As a model to characterize degradation, the flat film is acceptable. But physical properties are different between flat film and nanoparticle. So, the nanoparticles having multilayers is suggested to be evaluated.

4) In L. 223. According to Table 1., please change the order to “alginate, chondroitin, or hyaluronic acid”.

Author Response

The authors explained that the merits of cross-linking nanoparticles are to retard diffusion of drugs/proteins, and to improve degradation (e.g., P.1, L.34-38). In this paper, they demonstrated the degradation of their materials (nanoparticles and flat films), but they did not study the diffusion of drugs/proteins. The authors should add the drug release data.

The process of drug incorporation and release from the nanoparticles and multilayer system are a complex topic. First release studies of drugs from these particles and multilayers have been published already:

Cassan, D. de; Sydow, S.; Schmidt, N.; Behrens, P.; Roger, Y.; Hoffmann, A.; Glasmacher, B.; Menzel, H. Attachment of nanoparticulate drug-release systems on poly(ε-caprolactone) nanofibers via a graftpolymer as interlayer. Colloids Surf. B 2018, 163, 309–320.

Sydow, S.; Cassan, D. de; Hänsch, R.; Gengenbach, T.R.; Easton, C.D.; Thissen, H.; Menzel, H. Layer-by-layer deposition of chitosan nanoparticles as drug-release coatings for PCL nanofibers. Biomater. Sci. 2018, 7, 233–246.

Poth, N.; Seiffart, V.; Gross, G.; Menzel, H.; Dempwolf, W. Biodegradable chitosan nanoparticle coatings on titanium for the delivery of BMP-2. Biomolecules 2015, 5, 3–19.

Further release investigations have been carried out for BMP2 and TGF-ß3, which show that the release kinetics and efficiency for these signaling proteins depends on several parameters. This topic is part of intensive investigations and will be published in a separate paper, because a conclusive description would go beyond the scope of this paper.

 2) About nanoparticles. The authors used the CS-Az with 3.3 % Az groups. Can the authors supply more DLS data using the CS-AZ with different Az contents?

We included a figure for CS-Az nanoparticles with different degree of modification as supporting info. Increasing Az-content lead to increased particles size with a highly disperse particle size distribution. Furthermore, the stability of the particles decreased and zeta potential was not measureable. Multilayer construction with these nanoparticles failed. Therefore, we focused our investigations on the chitosan with 3.3% azide.  

3) The authors investigated degradation of CS-Az blocked multilayers prepared on silicon substrates. As a model to characterize degradation, the flat film is acceptable. But physical properties are different between flat film and nanoparticle. So, the nanoparticles having multilayers is suggested to be evaluated.

Nanoparticles made from chitosan/tripolyphosphate undergo a film forming process and transform to homogenous hydrogel film on a surface. The mechanism can be supposed to be first film formation as observed for water-based lacquers, that is polymeric particles are moving closer upon vaporization of the water and a subsequent coalescence of the particles occurs.

In a previous publication, we investigated the nanoparticle transforming process intensively by using XPS with depth profiling and proposed a mechanism for the film forming.

Sydow, S.; Cassan, D. de; Hänsch, R.; Gengenbach, T.R.; Easton, C.D.; Thissen, H.; Menzel, H. Layer-by-layer deposition of chitosan nanoparticles as drug-release coatings for PCL nanofibers. Biomater. Sci. 2018, 7, 233–246.

The final application of the CS/TPP system is a film on nanofibers, which is produced by the LbL-technique as described here, but only using electrospun fiber mats. (for details see: Cassan, D. de; Sydow, S.; Schmidt, N.; Behrens, P.; Roger, Y.; Hoffmann, A.; Glasmacher, B.; Menzel, H. Attachment of nanoparticulate drug-release systems on poly(ε-caprolactone) nanofibers via a graftpolymer as interlayer. Colloids Surf. B 2018, 163, 309–320.)

4) In L. 223. According to Table 1., please change the order to “alginate, chondroitin, or hyaluronic acid”.

Was corrected.

Reviewer 2 Report

The methods could be described better if you add a scheme of how the nanoparticles are synthesize as well as the chemical configuration of the molecules. Describing the possible arregement of the different layers, TPP interaction as well with the bioactive ingredients.

Also, even when the nanoparticles are described by size and charge, electron microscopy (SEM or TEM) is need it to corroborate the shape and size of the your system.

Author Response

The methods could be described better if you add a scheme of how the nanoparticles are synthesize as well as the chemical configuration of the molecules. Describing the possible arregement of the different layers, TPP interaction as well with the bioactive ingredients.

We included a scheme of the nanoparticle and multilayer preparation as well as the crosslinking of the azide layer. The methods should be clearer now. The (re)arrangement of the different layers and the interaction with TPP and the polyanions was studied in detail via XPS and published previously:

Sydow, S.; Cassan, D. de; Hänsch, R.; Gengenbach, T.R.; Easton, C.D.; Thissen, H.; Menzel, H. Layer-by-layer deposition of chitosan nanoparticles as drug-release coatings for PCL nanofibers. Biomater. Sci. 2018, 7, 233–246.

Also, even when the nanoparticles are described by size and charge, electron microscopy (SEM or TEM) is need it to corroborate the shape and size of the your system.

Indeed the particle size is investigated here only by dynamic light scattering and it would nice to have another method to investigate the particles. However, these nanosized hydrogels reorganize upon adsorption to a surface and lose their particle character. SEM measurements have been performed in a previous publication dealing with the general preparation of the multilayer systems:

Sydow, S.; Cassan, D. de; Hänsch, R.; Gengenbach, T.R.; Easton, C.D.; Thissen, H.; Menzel, H. Layer-by-layer deposition of chitosan nanoparticles as drug-release coatings for PCL nanofibers. Biomater. Sci. 2018, 7, 233–246.

We found, that nanosized hydrogel particles made from chitosan/tripolyphosphate undergo a film forming process and transform to homogenous hydrogel film on a surface. The mechanism can be supposed to be first film formation as observed for water-based lacquers, that is polymeric particles are moving closer upon vaporization of the water and a subsequent coalescence of the particles occurs. As a result, of this film forming process, no single particles are observable using SEM or TEM. In all attempts we saw only homogenous surfaces.

Even for single particles, removal of the water from the particle would result in a disruption of the particle character. Thus cryo-SEM would be the only possibility to observe the particles in solution, however, contrast is expected to be very low because of the high water content.

Round 2

Reviewer 2 Report

I received a file which hasn´t been change properly, but I did understand the changes.